# Electronic signatures of Lorentzian dynamics and charge fluctuations in lithiated graphite structures

Sasawat Jamnuch[1] & Tod A. Pascal [1,2] ✉

Lithium graphite intercalation compounds (Li-GICs) are essential materials for modern day portable electronics and obtaining insights into their atomic structure and thermodynamics is of fundamental interest. Here we explore the electronic and atomic states of Li-GICs at varying degrees of Lithium loading (i.e., "staging") by means of ab-initio molecular dynamics simulations and simulated X-ray adsorption spectroscopy (XAS). We analyze the atomic correlation functions and shows that the enhancements of the Li-ion entropy with increased staging result from Lorentzian lithium-ion dynamics and charge fluctuations, which activate low-energy phonon modes. The associated electronic signatures are modulations of the unoccupied $\pi^*/\sigma^*$ orbital energy levels and unambiguous fingerprints in Carbon K-edge XAS spectra. Thus, we extend the canonical view of XAS, establishing that these "static" measurements in fact encode the signature of the thermodynamic response and relaxation dynamics of the system. This causal link between atomic structure, spectroscopy, thermodynamics, and information theory can be generally exploited to better understand stability in solid-state electrochemical systems.

Commercially successful Lithium-ion batteries (LiBs) for renewable energy storage largely operate under the reversible intercalation of Li into a host material. Broadly speaking, Lithium intercalation provides a mean for controlled variation of the properties of the host material, including electrical, thermal, and even magnetic variations[1–5]. Modern LiB cathodes usually comprise transition metal oxides[6], with anodes comprised of graphite, both of which are integrated with composites, such as amorphous conductive carbon and polymer binders. During the electrochemical cycling of LiBs, the graphite anodes undergo reversible lithiation and delithiation, dynamically forming and collapsing lithium graphite intercalation compounds (Li-GICs) with various degrees of lithium loading[6]. These Li-GIC anodes present well-oriented, hexagonal carbon layers and anisotropic physico-chemical properties, such as a large in-plane conductivity (and negligible out-of-plane conductivity), which are further enhanced by ion intercalation[7]. Moreover, Li-ion intercalation between the graphite layers alters the structure from AB stacking into AA stacking, with -Li-$C_6$-Li-$C_6$- forming

a straight chain perpendicular to the sheet[8], eventually leading to an increase in the interlayer distance by 10%[9]. Thus, understanding the process and mechanism of lithium intercalation into graphite remains relevant, as model electrochemical systems.

Li-GICs with variable lithium loadings are referred to as being in different stages, with $LiC_6$ being stage I, $LiC_{12}$ as stage II, $LiC_{18}$ as stage III, and so on. Based on X-ray diffraction (XRD) studies, it has been established that the morphology of $LiC_6$ comprises Lithium atoms located in the center of adjacent carbon hexagons. Experimental measurements first suggested that the morphology of $LiC_{12}$ and $LiC_{18}$ is that of filled lithium layers and 1 or 2 empty layers respectively, as opposed to randomly distributed 1/2 and 1/3 filled layers[10]. This has been corroborated by extensive characterization studies, including XRD[11], angle-resolved X-ray emission spectroscopy (ARXES)[12], neutron diffraction[13], and X-ray Raman spectroscopy (XRS)[14]. While the exact thermodynamics driving force behind lithium staging is not well understood, it is commonly assumed that at lower lithium loading (i.e.,

[1]Department of Nano and Chemical Engineering, University of California San Diego, La Jolla, CA, USA. [2]Material Science and Engineering, University of California San Diego, La Jolla, CA, USA. ✉e-mail: tpascal@ucsd.edu

later stages), the Lithium entropy dominates the process, by increased configurational sampling, whereas at higher lithium loadings (earlier stages), enthalpy dominates[15].

In addition to the thermodynamic states, the electronic state of Li in Li-GICs, ranging from ionic to metallic, has been extensively studied[16–20], using a variety of modern experimental techniques[21–25]. Of particular interest are spectroscopic techniques, such as X-ray adsorption spectroscopy (XAS), which can directly probe the electronic states in the bulk, providing element-specific information about the local chemical environment with atomistic resolution. While the excitation energy of the Li K-edge (i.e., excitations from the 1s orbital to the conduction band) is ~56 eV and the penetration depth is rather shallow (10–20 nm), Carbon K-edge XAS has a penetration depth of ~100 nm, making it more suitable for in situ studies of battery electrode electronic structure and thus morphology. Both ARXES and XRS studies of highly oriented pyrolytic graphite (HOPG) and stage I fully lithiated HOPG (LiC$_6$) have been reported by Schulke et al.[22,23]. Boesenberg et al. used XRS to provide some more insight into the staging and mechanism of Li intercalation in Li-GICs[14]. At low momentum transfer, XRS is comparable to XAS. Fultz used electrochemical analysis to estimate the experimental entropy of Li intercalation into graphite and concluded that the entropy is dominated by configurational entropy at low Li concentration while at higher concentration it is dominated by vibrational contributions[15].

While much progress has been made over the years, the underlying physics behind Li-GICs has not been fully elaborated, nor have the atomistic dynamics that give rise to specific X-ray spectral features been elucidated. This is a critical knowledge gap since future rational design strategies for improved battery electrode materials will undoubtedly depend on an appreciation of the complex quantum mechanical effects that underlie the atomic scale morphology. Here we quantify the role of Lithium-ion dynamics on the associated XAS spectral features computationally, using extensive, ab-initio molecular dynamics (AIMD) simulations, entropy quantification from information theory and core-level XAS calculations employing a many-body, constrained occupancy density functional theory (DFT) formulism[26,27]. We show that excellent agreement between our simulated XAS spectra and experimental measurements can be achieved at this level of

theory, for graphite and at all levels of staging. Further, we show that the experimentally observed differences in the XAS spectra with staging are in-fact a fingerprint of charge transfer fluctuations in the Li – graphite networks and non-Gaussian, finite temperature Li – ion dynamics, that populate long wavelength, correlated vibrational modes. These vibrational modes are then shown to be critical for accurately accounting for the system thermodynamics, beyond what can be predicted from a purely harmonic theory, therefore resolving some of the disagreement between previous calculations and experiments that exists in the literature. Beyond further establishing XAS as a useful technique for probing the local chemical environment in material systems, this study demonstrates the power of simulated XAS and spectral analysis in extracting dynamical information, and even in quantifying the thermodynamics, of functional materials, while highlighting the important role of non-Gaussian dynamics in modulating the entropy and informing overall system stability.

## Results and discussion

The crystal structures of graphite and stage I–III Li-GICs used in this study are shown in Fig. 1a. For each system, we performed AIMD simulations at 298 K and used 5 snapshots to simulate the XAS in Fig. 1b. Previous studies have shown that instantaneous distortions brought on by finite temperature thermal effects are necessary for reproducing the experimental XAS in various lithium compounds[28]. Moreover, we rationalize that since the timescale of electronic response (attoseconds) is significantly shorter than that of the motion of the nuclei (femtoseconds), the experimentally observed x-ray spectroscopy is in fact the statistical ensemble averaged spectra resulting from many such snapshots. We then simulated the XAS spectra at the C K-edge by statistical averaging over all the carbon atoms in each snapshot. Of note, the C K-edge XAS of graphite is characterized by two prominent features, a peak near 285.5 eV due to excitations from the 1s orbital to π*-resonances and a less intense peak near 291.7 eV due to 1s → σ*-resonances. At the σ* resonance, the broadening of this feature is observed in the experimental spectrum as staging progresses[14].

Overall, the XAS simulated from sampling our AIMD trajectory was found to be in excellent agreement with the experiment, for

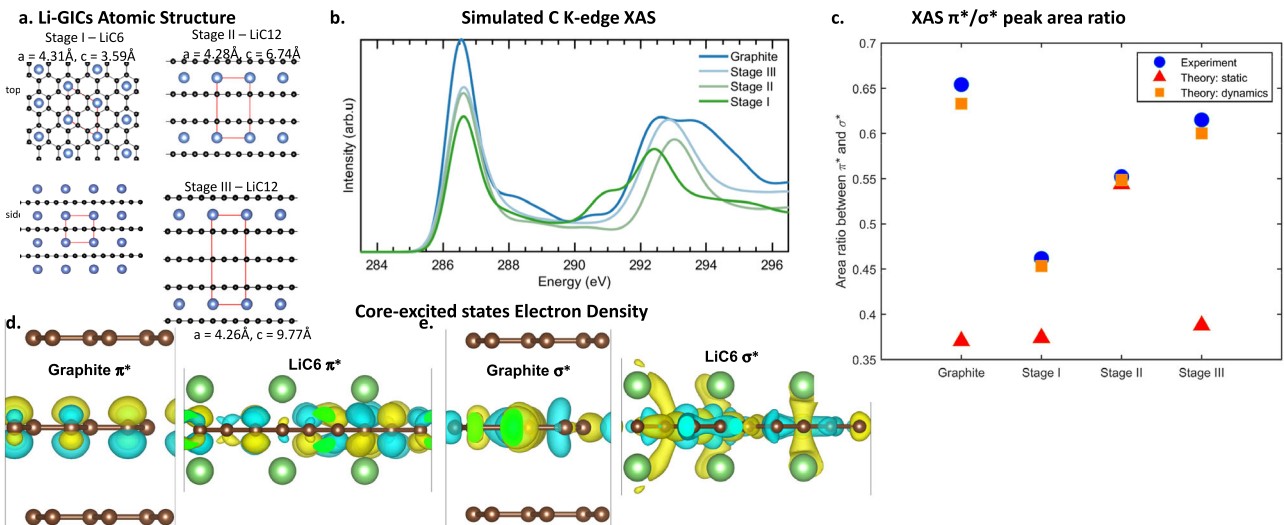

**Fig. 1 | Electronic structure of graphite and staged Li-GICs. a** Schematic showing Li staging of graphite. **b** Simulated C K-edge of pristine graphite and all three Li-GICs, using samples from AIMD trajectories. **c** Comparison of the π*/σ* peak area ratio for the various Li-GICs, from experiment[14] (blue circles), simulations of the static 0 K optimized structure (red triangles) and simulations sampled from a 298 K AIMD trajectory (orange squares). **d** Visualization of the π* core-excited state

electron densities in graphite (left) and LiC$_6$ (right). **e** C atoms are in brown and Li atoms are shown in green. We adopt the convention where the positive phase of the electron density is shown in yellow while the negative phase is shown in blue. Visualization of the σ* core-excited state electron densities. **e** Visualization of the σ* core-excited state electron densities in graphite (left) and LiC$_6$ (right). Source data are provided as a Source Data file.

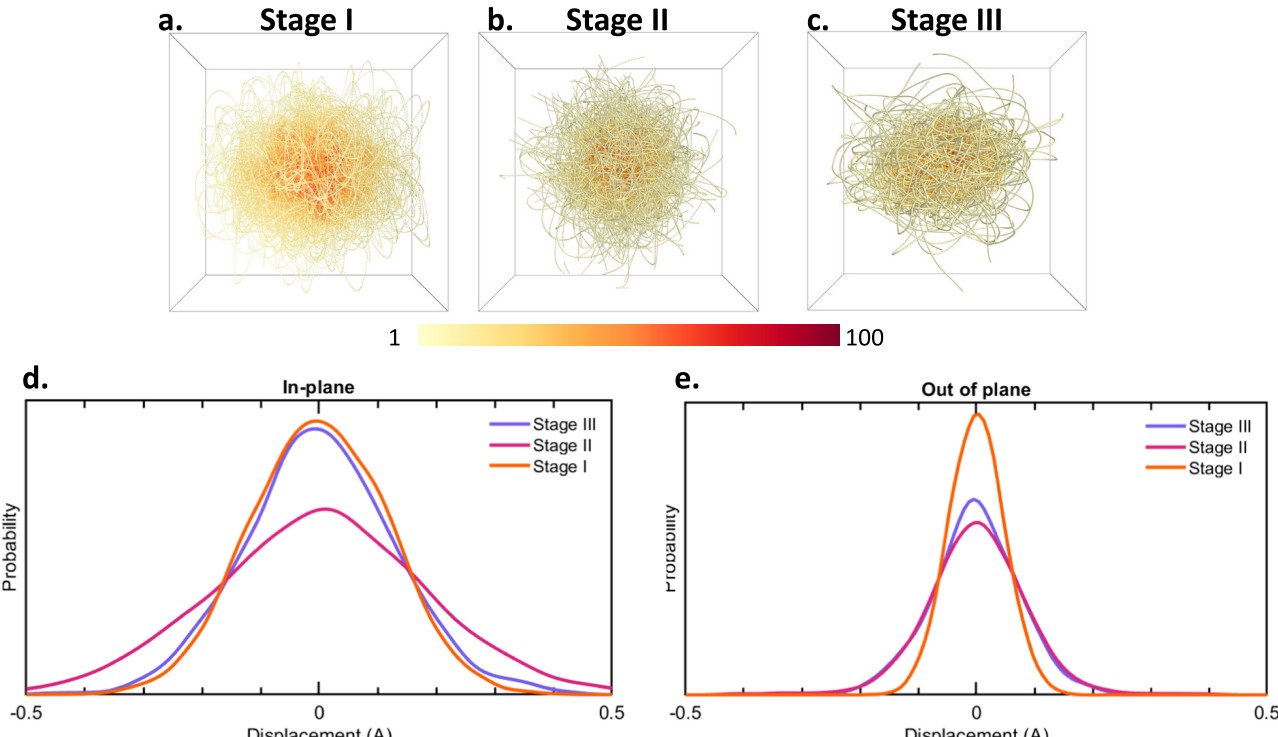

**Fig. 2 | Structural analysis of Li-GICs. a–c** Li atom librational displacement within the graphitic cage, in LiC$_6$, LiC$_{12}$, and LiC$_{18}$ respectively, showing the probability weight of lithium position within the graphitic cage. **d** *In-plane* projection of the Li librational cage displacement dynamics. **e** *Out-of-plane* projection of the Li librational cage displacement dynamics. Source data are provided as a Source Data file.

graphite and all three Li-GICs, both in peak positions and lineshape (Fig. 1b). On the other hand, the simulated XAS using the 0 K optimized, static structure (Supplementary Fig. 2) was found to be in poorer agreement with experiment, underscoring the role of atomic fluctuations in modulating the XAS. This is especially notable at the σ* resonance, where simulations of the static structure greatly overestimated the oscillator strength. To quantify this effect, we fitted each feature with a Lorentzian function and integrated the area under the curve. The resulting π*/σ* peak area ratios are shown in Fig. 1c, with a correlation coefficient of $R^2 = 0.97$ for the finite temperature XAS compared to experiments, but only $R^2 = 0.13$ for the static structure. Details of our fitting procedure and parameters can be found in Supplementary Methods, Supplementary Fig. 1 and Supplementary Table 1.

Insights into the modulation of the XAS during staging were further obtained by constructing hypothetical models with controlled variations, such as stacking orientation and interlayer distance. In graphite, the stacking is AB with a layer separation distance of 3.35 Å, while Li-GICs generally have AA stacking with a layer separation distance of ~3.7 Å. Consider first the XAS of graphite. We find that the Carbon K-edge XAS is relatively insensitive to AA/AB stacking (Supplementary Fig. 3) or interlayer distance in the AA stacked structure (Supplementary Fig. 4), suggesting that the weak sheet-sheet interactions manifest out-of-plane π* features (Fig. 1d, e) that are similarly uncoupled. The in-plane σ* peaks are found to be more sensitive to bond length changes, such as those that would be expected from finite temperature thermal fluctuations. Indeed, as shown in Supplementary Fig. 5, we find significant modulation in the position of the σ* peak with varying in-plane lattice constant. As the in-plane bonds contracts, the σ* peak position blue shifts due to increased splitting between bonding and antibonding valence molecular orbitals, resulting from the increased spatial overlap of atomic orbitals[29]. Therefore, the broadness of the σ* peak reflects the extent of in-plane C–C bond fluctuations in

graphite at finite temperature. In Supplementary Fig. 6 we find shorter C–C bonds but a smaller degree of fluctuation, in graphite (1.42 ± 0.02), compared to LiC$_6$ (1.44 ± 0.04), somewhat explaining the broader σ* XAS peak (and reduced peak area ratio) in LiC$_6$.

Figure 2a–c plots the Li-ion displacement trajectory for the various Li-GICs from AIMD simulations at 298 K, in the graphite cage defined instantaneously by the 12 nearest carbon atoms (6 top and 6 bottom). Supplementary Table 2 details an analysis projecting the Li-ion cage displacement along the in-plane and out-of-plane axis, based on the calculated higher-order moments (i.e., the skewness and kurtosis) compared to a normal distribution of the same variance. While the dynamics of the Li ions are symmetric about the graphite cage center of mass, we find that the distribution of Li ions have large Kurtosis and are in fact largely Lorentzian. We also analyzed the angle correlation between the center of carbon of the first hexagon, Li atom position and center of carbon of the second hexagon, where we find a similar non-Gaussian distribution (Supplementary Fig. 7). The underlying graphite cage atomic microstructure is the origin of this effect, i.e., the anisotropic potential in the graphite cage[30] greatly modulates the Li-ion in-plane diffusion, such that the Li ions have a higher than expected probability of residing at the midpoint of the C–C bonds. This leads to longer tails in the distribution. Further, we calculated that the distributions become more Lorentzian-like with increased staging, due to larger fluctuations in the C–C bond distributions, which in turn leads to increased broadening of the σ* peak in the XAS. Supplementary Table 3 details the charge distribution analysis for the various Li-GICs, showing increased Lorentzian distribution with staging, due to Li-ion dynamics. Various previous works have utilized structural probes to track the motion of lithium ions in electrode materials during charge and discharge[31–33]. We propose that the Lorentzian ion distributions, in- and out-of-plane with staging, can be verified by these complementary techniques, such as high-resolution neutron scattering.

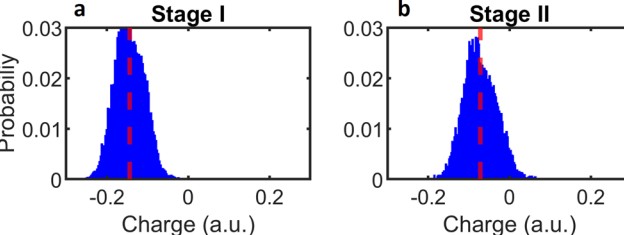
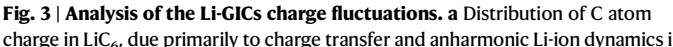
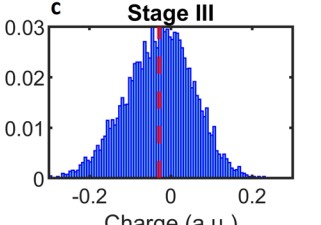
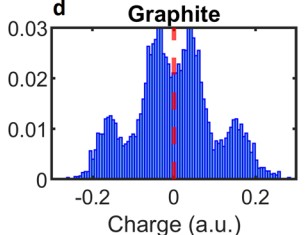

**Fig. 3 | Analysis of the Li-GICs charge fluctuations. a** Distribution of C atom charge in LiC$_6$, due primarily to charge transfer and anharmonic Li-ion dynamics in the graphite cage (*left inset*), **b** LiC$_{12}$ and **c** LiC$_{18}$, and **d** graphite from charge puddling. Source data are provided as a Source Data file.

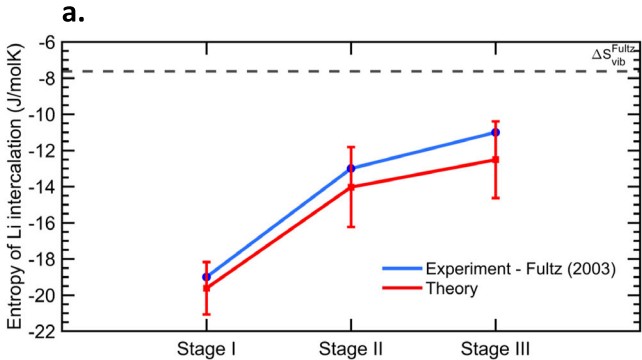
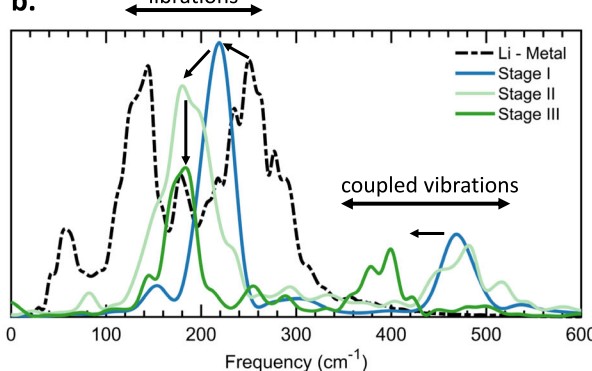

**Fig. 4 | Thermodynamics of staged Li-GICs. a** The entropy of Li intercalation into Li-GICs, relative to Lithium metal. Our calculated values from 298 K AIMD simulation (red) are compared to the experiments (blue). The error bars on the calculated values represent the uncertainty (standard deviation 1σ). The calculated entropy change assuming purely harmonic vibrations is shown by the vertical dashed line. **b** Comparison of the spectral vibrational density of states (i.e., power spectrum) of Lithium in the bulk metal (dashed black line) and in LiC$_6$ (blue), LiC$_{12}$ (green) and LiC$_{18}$ (red). Source data are provided as a Source Data file.

Lorentzian dynamics in Li-GICs induces secondary charge transfer and fluctuation physics that also modulates the XAS peak positions, and thus the relative intensity of the σ* resonance. Figure 3 shows the distribution of charges on the carbon atoms as a function of staging. We note that the seemingly unexpected charge distribution in HOPG graphite is due to intrinsic charge fluctuations due to the sheet vibrations and the associated electron–hole puddling physics[34,35]. For the Li-GICs, we find charge transfer from the Lithium atoms into graphite, such that the formally Li$^0$ in the metal becomes Li$^{+0.8}$, Li$^{+0.7}$, and Li$^{+0.6}$ in stages I, II, and III respectively. Li et al. also demonstrated a similar degree of charge transfer in Li-GICs arising from the dominant interaction between Li 2s and C 2p$_z$[36], while Maher et al. showed that experimentally a fully discharged state cell is still partly charged[37]. The increased electron density around the carbon atoms in Li-GICs leads to a reduced binding energy of the 1s core–electrons and an overall redshift of the spectrum, consistent with the energy changes in the π* feature with increasing staging. This in turn modifies the electronic structure around the Fermi level and suppresses the intensity of the π* feature due to state blocking[38]. Our results are consistent with a previous electronic structure calculation[17] which showed that LiC$_6$ conduction bands arise from the interaction between Li 2s and C π electrons.

Given the charge transfer and dynamics of Li ions in graphite, we further found that modulation of the XAS spectra of graphite and Li-GICs, as a function of staging, is a measure of the overall Lithium-ion thermodynamics. To quantify this, we calculated the total Li entropy, using our 298 K AIMD trajectory and a statistical approach based on information theory, where the entropy is calculated from the velocity autocorrelation function[39,40]. Critically, this approach explicitly includes both harmonic and anharmonic effects. We found that the Li entropy is lower in Li-GICs than in Lithium

metal, with −19.62 J/mol/K for LiC$_6$, −14.03 J/mol/K for LiC$_{12}$ and −12.51 J/mol/K for LiC$_{18}$ (Fig. 4a). These values are in excellent agreement with experimental results (−19, −13 and −11 respectively), and directly results from the non-Gaussian/anharmonic Li-ion dynamics detailed earlier. Indeed, assuming purely harmonic contributions to the vibrational entropy would lead to a constant value of ΔS = −7.6 J/mol/K for all stages[15].

We found that the modulation in the Li-ion entropy and thermodynamics with staging, in turn, has unique vibrational spectroscopic fingerprints. Figure 4b shows that Li intercalation into graphite suppresses low energy (long wavelength) modes at 80 cm$^{-1}$ and 120 cm$^{-1}$, which are present in the metal due to Li ↔ Li librations. Additionally, we calculated that a Li rattling mode at 250 cm$^{-1}$ in the metal blue shifts to 200 cm$^{-1}$ (stage I) and to 180 cm$^{-1}$ (stage II & III), reflecting the overall softer potential experienced by Li-ion in Li-GICs. This softer lithium potential in Stage II and III is due to reduced Li-Li repulsion, in agreement a previous DFT study[20]. We also found a new feature at 470 cm$^{-1}$ (stage I and II) and 400 cm$^{-1}$ (stage III), which is the signature of charge transfer to the graphite host. This mode has been found to be Raman active mode arising from the coupling between intercalated ions and graphite in-plane motion[41]. Finally, we re-emphasizes how critical it is to include anharmonic vibrational contributions when calculating the entropy of Li in LiGICs (and we suspect other intercalation compounds). Indeed, the role of dynamics on the "configurational entropy" has recently been proposed as critical for understanding the cathode materials in Lithium-ion batteries[42].

In summary, we have established a causal link between the energy and intensity in the XAS spectra of Li-GICs, and the Li ion dynamics and thermodynamics. In doing so, we show that "static" XAS measurements in fact encode subtle and complex microscopic physics, using Lorentzian dynamics. More generally, the presence of these

anharmonic entropy contributions may present new avenues whereby the thermodynamic states in these materials can be manipulated, for example by selective phonon activation to populate or depopulate certain vibrational modes, which may provide an alternative explanation for recent observations such as the increased ion transport in batteries by light illumination[43]. To this end, we view the computational approach presented here as broadly applicable to more complex intercalation electrode chemistries, as a general way of understanding how microscopic structure and dynamics determine electrochemical stability and function.

## Methods

### Molecular dynamics simulation

Ab-initio DFT calculations, using the projector augmented wave method implemented in VASP[44], were used to simulate the finite temperature (298 K) dynamics of graphite and Li-GICs. We employed the Perdew–Burke–Ernzerhof exchange-correlation functional[45], along with DFT-D3[46] van de Waal corrections. The systems were sampled using a Monkhrost-Pack[47] $12 \times 12 \times 12$ k-point mesh for Brillouin zone integration. We initially minimized the systems at 0 K to a force tolerance of 0.01 eV/Å. The optimized supercell parameters are shown in Supplementary Table 4. Next, molecular dynamics (AIMD) simulations were performed, using a $3 \times 3 \times 3$ supercell, obtained from the minimized structures, which allowed us to decrease the Brillouin zone sampling to a $2 \times 2 \times 2$ k-point mesh, increasing computational efficiency. The supercells consisted of 108 atoms for the graphite system and 189, 234, and 171 for $LiC_{6-18}$ respectively. We used large supercells so that the cell boundaries were all greater than 10 Å, necessary to accommodate the core-excited states in our XAS calculations, while also reducing the self-interaction between atoms due to the effect of artificial periodicity. We propagated the system forward in time with a numerical integration timestep of 0.5 fs, using the velocity verlet algorithm. The temperature of the system was kept near 298 K by means of a Nose thermostat (time sampling constant of 20 fs). We resolved the temperature-related stress in the system by means of a 2 ps constant temperature, constant pressure (NPT) simulation. This was then followed by at least 10 ps of constant temperature, and constant volume (NVT) dynamics. During the final 5 ps of the NVT simulation, snapshots of the system (atomic positions and velocities) were saved every 5 fs.

### Partial atomic charge analysis

The charges on each individual atom were calculated from the electron charge density obtained from VASP electronic ground state calculations. We used 50 uncorrelated snapshots, extracted from the MD trajectories [evenly spaced from the last 5 ps of our MD simulation], and the partial atomic charges determined by the Bader charge analysis method[48]. We used a Brillouin zone integration with a coarser $2 \times 2 \times 2$ k-point grid for computational efficiency, but verified that this produced nearly identical results to a denser $12 \times 12 \times 12$ grid (Supplementary Fig. 8).

### Carbon K-edge XAS calculations

We simulated the Carbon K-edge XAS of graphite and Li-GICs using a many-body ΔSCF approach, ShirleyXAS[49] + MBXASPY[27]. The final electronic state due to the x-ray excitation was obtained using the full core-hole approach, where the Carbon 1 s electron was removed from the inner shell. This was achieved by modifying the carbon pseudopotential to generate a $1s^1$ core-hole, where both the ground-state and core-excited state pseudopotentials were prepared using the ultrasoft Vanderbilt pseudopotential scheme[50]. The kinetic energy cutoff was 30 Ryd and the energy cutoff for charge density was 240 Ryd. Next, we performed non-self-consistent field calculation (NSCF), a single shot calculation to construct the Hamiltonian without updating the charge density, to access empty states at higher energy. Empty bands with at least double the amount of occupied bands were included in the NSCF calculation. The convergences of XAS simulation as a function of supercell size or number of empty bands are shown in Supplementary Fig. 9-10, where we find that the convergence is achieved at $3 \times 3 \times 3$ unit cell with a minimum band factor of 3. The XAS spectrum was obtained by applying Fermi's golden rule, where the oscillator strength and overlap between the initial (ground) state and the core-excited states were obtained from the Slater determinant. Each simulated spectrum was broadened by a Gaussian smoothening of 0.1 eV and a rigid shift of 285.5 eV was applied to each spectrum, based on a reference calculation where we aligned the Carbon K-edge of molecular CO to experiment. Further energy calibration was performed according to the formation energy scheme[28,51]. This allowed us to calculate accurate energy shifts between the various Li-GICs and thus compare the spectra of systems with different chemical environments. To overcome the underestimation of the band gap (and concomitantly the bandwidth) at this level of DFT, we applied an empirical dilation of 5% to each spectrum to match the splitting between the π* and σ* features in the experimental graphite spectrum, and this same dilation factor is applied to all Li-GICs spectra. For each MD snapshot, we simulated the XAS of each carbon atom individually, and the final spectra were obtained from statistical averaging. The reported results were obtained by averaging 5 uncorrelated snapshots over 5 ps, each 1 ps apart. The uncertainty in the simulated spectra due to statistical averaging is shown in Supplementary Fig. 11. For comparison to experiments, we matched the π* peak intensity in graphite to that of our simulated XAS of graphite and used the same scaling factors for all other simulated spectra. This was necessary due to the inherent differences in the normalization methods between the experiment and simulation: in the experiment, the spectra are normalized based on background signal subtraction whereas the calculation does not have any background response. Our approach thus allows for quantitative comparison across staging.

### Evaluation of thermodynamic properties

The thermodynamic properties of Li-GICs were calculated using the Two-Phase Thermodynamic method, using the trajectories from the last 5 ps of our AIMD simulations. We first calculated the density of states (DoS, also known as the spectral density) function as a Fourier transform of the atomic velocity autocorrelation function. We then calculated the entropy by separately considering the diffusive and vibrational motions of the atoms. Notably, since the DoS is obtained directly from MD, the resulting entropy inherently contains contributions due to harmonic (purely vibrational), as well as anharmonic (librational) and self-diffusive motions. We find that the latter two are the dominant contributions to the system Li entropy.

### Reporting summary

Further information on research design is available in the Nature Portfolio Reporting Summary linked to this article.

## Data availability

The authors confirm that all relevant data are included in the paper and/or its supplementary information files. Source data for all figures, atomic structures, and VASP input files used for the various calculations have been deposited in https://github.com/atlas-nano/lithiated_graphite_xas [https://doi.org/10.5281/zenodo.7739094].

## Code availability

The thermodynamics were obtained by post-trajectory analysis of a code that implements the 2PT method. This code can be accessed from our Github repository: https://github.com/atlas-nano/2PT [https://doi.org/10.5281/zenodo.7731073]. Source data are provided in this paper.

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

## Acknowledgements

This research was supported by the U.S. Department of Energy (DOE), Office of Basic Energy Science, Grant No. DE-SC0023503. S.J. acknowledges funding from the UC Office of the President within the Multicampus Research Programs and Initiatives (M21PL3263). This work used the Extreme Science and Engineering Discovery Environment (XSEDE) resources on the EXPANSE supercomputer at the San Diego Super Computing Center (SDSC). Portions of this work were completed as a user project by S.J. at the Molecular Foundry, a US Department of Energy Nanoscience Facility at Lawrence Berkeley National Laboratory supported by the Office of Science, Office of Basic Energy Sciences, of the U.S. DOE under Contract No. DE-AC02-05CH11231. Some simulations used resources from the National Energy Research Scientific Computing Center, which is supported by the Office of Science of the U.S. DOE under the same contract.

## Author contributions

Correspondence should be addressed to T.A.P. (tpascal@ucsd.edu). T.A.P. conceived the study and supervised the project. S.J. performed the MD and XAS simulations, and the thermodynamics calculations. Both authors contributed to the analysis, discussion of the data, and writing of the manuscript.

## Competing interests

The authors declare no competing interests.
