## [Peer Review File · Nature Communications]

Electronic signatures of Lorentzian dynamics and charge fluctuations in lithiated graphite structuresREVIEWER COMMENTS

Reviewer #1 (Remarks to the Author):

The manuscript by Jamnuch and Pascal reports the interpretation of X-ray absorption spectra for the Li-graphite system based on first-principles calculations. The authors find that the spectra are strongly temperature-dependent, and simulations at room temperature were needed to reproduce features seen in experiment. Graphite is a common negative electrode material in Li-ion batteries, and so the topic is of wide interest. The approach seems technically sound, and the manuscript is mostly well written. Though, the subject, temperature dependence of XAS, is rather technical and a more specialized journal might perhaps be more appropriate.

There are a few issues that the authors should address in a revision before I can recommend publication:

1. The introduction section makes it sound as if graphite was a common cathode (positive electrode) material for LIBs. This is incorrect. However, graphite-based materials are the most commonly used anode (negative electrode) in commercial LIBs. In fact, Akira Yoshino received 1/3 of the 2019 Nobel Prize in chemistry for this development. The introduction section should be revised accordingly.

2. Previous computational work by Holzwarth and by Persson (cited in the present manuscript) already established a good understanding of the thermodynamics and Li diffusion kinetics of the Li-graphite system. This is, in my opinion, currently misrepresented in the introduction. Notably, the work by Persson also accounted for temperature effects, but only for configurational entropy and not for vibrational entropy. Are the findings in the present work compatible with the previous work by Persson? Can vibrational entropy contributions explain discrepancies between experiment and simulation that were observed in the older publications? Please include a discussion in the revised manuscript.

3. The methods section lacks many important details that are required for reproducibility. It is stated that the "systems" were sampled with a 12x12x12 k-point mesh. What unit cells did this apply to (structure, composition, and number of atoms)? It seems, the k-point density was reduced for AIMD simulations. How did this affect the accuracy of the atomic forces (which are crucial for the vibrational entropy)? What was the time step of the AIMD simulations? Was the temperature-dependent volume change of the unit cell accounted for? Precisely which pseudopotentials were used, especially for the XCH calculations? How many empty bands were included in the XCH calculations, and was this parameter converged/benchmarked? Was the full-core-hole approximation used? Were the spectra post-processed, e.g., were they broadened or shifted?

4. The manuscripts contains several typos that a spell-/grammar-checker should have detected. Note that the name "Monkhorst" is misspelled.

Reviewer #2 (Remarks to the Author):

This is a short, interesting paper that reports on the impact of lattice and ion dynamics in lithium intercalated graphite. The authors calculate the carbon K-edge X-ray absorption spectra (XAS) and compare their results to experiment and conclude that the vibrational properties have a strong impact on the thermodynamics of lithium intercalation. Well interesting, this paper is not significantly important to the field and there are some technical limitations that prevent its publication.

The introduction is quite long relative to the length of the body of the paper. It seems that much in the introduction is unnecessary. In addition, the first paragraph of the introduction states that graphite can be used as a cathode and in fact uses this motivation throughout the rest of the paper. This is highly problematic as graphite in all lithium-ion batteries are anodes. The potential

of lithiated graphite is very low potential (about 0.1 volts versus Li metal) and this would render any battery with graphite as a cathode basically useless in terms of energy density. The introduction needs to be condensed and corrected. In addition, the importance of the results are not adequately explained. For example, what is the impact of these on lithium intercalation on battery operation. The discussion in the conclusion related to phonon activation with electromagnetic fields seems highly unlikely given that lithium ion batteries all have metal casings which are required for current conduction from the electrodes to the external circuit.

The authors make strong statements about their room-temperature XAS results agreeing with the experiment as opposed to the 0K results (no lattice dynamics); this is based on figure 1C which just looks at the peak areas and may not be representative of the complete XAS spectra. The authors need to directly compare the calculated XAS with those of the experiments in order to show that their simulations provide more accurate results. Similar comments apply to the Raman results in Figure 4. These comparisons are essential.

On Page 3 in the second full paragraph, the authors state that there is incomplete charge transfer to the lithium ions and that the formal charge on the lithium is +0.8, +0.7 and +0.6 for stages I to III. It is unclear how this could possibly agree with the experiment which largely sees complete charge transfer to the Li and is the basis for the operation of Li-ion batteries (i.e., charge capacity). These results need to be rationalized and explained.

The last paragraph on page 2 discusses what aspects of the intercalated graphite structure give rise to the XAS features. The conclusions for this are obvious as XAS is a highly local probe and hence will only see the ordering within the planes (short bond lengths) and will be insensitive to the interactions between the graphite planes (longer distances). This paragraph adds very little to the paper that should be a move to the SI.

Reviewer #3 (Remarks to the Author):

The authors present a calculation-driven investigation of the motion of Li atoms intercalated into graphite, a system used as a cathode in lithium ion batteries. Not unexpectedly, they show that the lithium atoms move a fair bit, but more interestingly they also show that the distribution is very far from Gaussian (not reflective of a harmonic potential) and that this motion and its non-harmonic behavior has a dramatic effect on the enthalpy of the intercalated system. The predicted dynamics of the Li-graphite systems are fed in to first-principles calculations of the C K-edge x-ray absorption. As a point of comparison with experimental measurements, the peak ratio between the two largest features in the XAS are compared as a function of Li concentration (staging), and the authors find very good agreement between the predictions and measurements for this ratio.

I have a few major concerns. The comparison with experiment is limited to the ratio of the two main peaks in the C K edge. As the current paper is written, the authors hinge much of their arguments on the agreement between experiment and calculation of this ratio, but I do not think it is adequately supported.

1. Peak ratios are going to be sensitive to the details of the fitting, such as the determination of the background level and bounds of the peaks. The method needs to be clearly laid out in either the paper or the supplemental. I would suggest a supplemental figure illustrating this for both the calculated and measured spectra. It is not clear which experimental data was used for generating the comparison in Fig 1c, but it should be cited in the caption and in the text where Fig 1c is mentioned.

2. Are the qualitative changes in the calculated spectra that are evident in Fig 1b borne out in experiment? Or changes in the onset energy of the sigma star?

I have no major criticisms with the computational methodology used. In a few cases the authors need to be more clear. In particular:

3. What were the volumes used for each of the 4 systems? Was this from experiment or from relaxing with respect to the pressure of the unit cell? If the latter, how does this compare with measured data, and if the former, what was the pressure from the DFT?

4. How many snapshots went into the XAS calculations? (you state the corresponding numbers for the partial charge analysis and vibration/entropy parts.) What is meant by "statistical averaging" for generating the final spectra?

Additional comments:

Stylistically, the authors may want to put more weight into their finding that the Li entropy depends strongly on the anharmonic effects and the motion of the graphite sheets. The first full paragraph on page 2 mentions the entropy calculations, but goes on to only summarize XAS results. Along these lines, I think more emphasis could be placed on the important role that non-Gaussian dynamics can have on a system's thermodynamics. The authors cover this well in the final paragraph before the methods, but it could be brought up in the introductory remarks as well.

The manuscript makes a distinction between XAS and XRS, but for low momentum transfer XRS can provide the same information. That is to say that, Ref 15 provides information directly comparable to the XAS calculations, and the techniques used here are perfectly fine for simulating low-q XRS.

The authors link the XAS to the motion of the Li through their AIMD. Would structural probes like neutrons be able to also confirm the level of in-plane and out-of-plane displacement (though perhaps limited to modeling only the width of the distribution)?

Fig 2a-c should be described better in the caption — what the color bar represents.

In Fig 2d / AIMD trajectories are there any meaningful angle correlations for the Li motion with respect to the C hexagons?

Optional, but would it be feasible to also include calculations of the vibrational DOS from within an entirely harmonic approximation? This might give an interesting point of comparison for harmonic vs anharmonic motion.

RESPONSE TO REVIEWERS' COMMENTS

Reviewer #1 (Remarks to the Author):

We thank the reviewer for their review of our work. Your comments were very helpful in improving the overall quality the manuscript. We address your concerns below. Your comments are reproduced verbatim in *italics* while our response follows in red. Where appropriate, we will indicate any major changes to the manuscript or SI by underline.

The manuscript by Jamnuch and Pascal reports the interpretation of X-ray absorption spectra for the Li-graphite system based on first-principles calculations. The authors find that the spectra are strongly temperature-dependent, and simulations at room temperature were needed to reproduce features seen in experiment. Graphite is a common negative electrode material in Li-ion batteries, and so the topic is of wide interest. The approach seems technically sound, and the manuscript is mostly well written. Though, the subject, temperature dependence of XAS, is rather technical and a more specialized journal might perhaps be more appropriate.

There are a few issues that the authors should address in a revision before I can recommend publication:

1. The introduction section makes it sound as if graphite was a common cathode (positive electrode) material for LIBs. This is incorrect. However, graphite-based materials are the most commonly used anode (negative electrode) in commercial LIBs. In fact, Akira Yoshino received 1/3 of the 2019 Nobel Prize in chemistry for this development. The introduction section should be revised accordingly.

We thank the reviewer for the comment pointing out this typo. The introduction has been revised accordingly. The introduction now reads:

“... and even magnetic variations.¹⁻⁵ Modern cathodes usually comprise transition metal oxides,⁶ with anodes comprised of, most recently, Lithium metal. However, anodes of highly oriented pyrolytic Graphite (HOPG), operating under the principle of reversible intercalation of graphite during charge and discharge and otherwise known as Lithium graphite intercalation compounds (Li-GICs), are also widely used.⁶ Thus, understanding the process of lithium intercalation into graphite remains relevant, as model electrode materials.”

2. Previous computational work by Holzwarth and by Persson (cited in the present manuscript) already established a good understanding of the thermodynamics and Li diffusion kinetics of the Li-graphite system. This is, in my opinion, currently misrepresented in the introduction. Notably, the work by Persson also accounted for temperature effects, but only for configurational entropy and not for vibrational entropy. Are the findings in the present work compatible with the previous work by Persson?

We thank the reviewer for the comment. Our current results expand on these previous works, to now explicitly account for the effect of temperature. Indeed, as alluded to by the reviewer, one of the key findings of our work is the role of vibrational entropy in determining the overall system thermodynamics. We added the following discussion points to the manuscript:

“...from state blocking due to intercalation.³⁴ This later result is in general agreement with previous

electronic structure calculations¹⁵ which showed that the LiC₆ conduction bands arise from interactions between Li 2s and C π electrons. Table S4 details the charge distribution ...

and

“...reflecting the softer potential experienced by Li-ion. A previous DFT work¹⁸ also calculated softer Li potentials in Stage II and III due to reduced Li-Li repulsion.”

Can vibrational entropy contributions explain discrepancies between experiment and simulation that were observed in the older publications? Please include a discussion in the revised manuscript.

We believe that inclusion of vibrational entropy is essential for capturing the thermodynamics of the system. This was pointed out in the original paper of Yazami and Fultz [Journal of Power Sources 119-121, 850-855, doi:10.1016/s0378-7753(03)00285-4 (2003)]. A more recent work has also shown the need to include vibrational entropy when determining the thermodynamics of Lithium ions in complex cathode materials [Friedrich et. al., J ECS 168, 120502, doi:10.1149/1945-7111/ac3938 (2021)]. We expand on these works to show that one needs to further go beyond the purely harmonic approximation in order to fully appreciate the physics as shown in Fig. 4a. We have modified the manuscript to now read:

“We found lower Li entropy in Li-GICs than in Lithium metal: $\Delta S = -19.6, -14.0$ and -12.5 J/mol/K for LiC₆, LiC₁₂ and LiC₁₈ respectively (Fig. 4a). These values are in excellent agreement with the results from electrochemical experiments ($-19, -13$ and -11 J/mol/K respectively).¹³ Most importantly, these entropic effects results directly from anharmonic Li-ion dynamics. Indeed, assuming purely harmonic contributions to the vibrational entropy leads to a constant value of $\Delta S = -7.6$ J/mol/K for all stages.¹³”

3. The methods section lacks many important details that are required for reproducibility. It is stated that the "systems" were sampled with a 12x12x12 k-point mesh. What unit cells did this apply to (structure, composition, and number of atoms)?

We thank the reviewer for the comment. The method section has been fully updated to include details of all calculations. Specifically, the supercells consisted of 108 atoms for graphite system and 189, 234 and 171 for LiC₆₋₁₈ respectively.

It seems, the k-point density was reduced for AIMD simulations. How did this affect the accuracy of the atomic forces (which are crucial for the vibrational entropy)?

The AIMD used a 3x3x3 supercell and a 2x2x2 k-point grid. We repeated our energy minimization using these same parameters and found no difference in the forces compared to using the unit cell and a 12x12x12 k-point grid. We added the following clarifying sentence in the method section:

“We used large supercells so that the cell boundaries were all greater than 10 Å, necessary to accommodate the core-excited states in our XAS calculations, while also reducing the self-interaction between atoms. Additionally, this allowed us to perform the simulation at lower k-point mesh, increasing computational efficiency.”

What was the time step of the AIMD simulations?

We used a 0.5fs integration timestep

Was the temperature-dependent volume change of the unit cell accounted for?

We performed an initial 2ps constant temperature, constant pressure (NPT) calculation to equilibrate the temperature. The modified method section now reads:

“The temperature of the system was kept near 298K by means of a Nose thermostat (time sampling constant of 20 fs). We resolved the temperature related stress in the system by means of a 2ps constant temperature, constant pressure (NPT) simulation. This was then followed by at least 10 ps of constant temperature, constant volume (NVT) dynamics.”

Precisely which pseudopotentials were used, especially for the XCH calculations?

Both the ground-state and core-excited states used ultrasoft pseudopotentials, in the case of the core-excited states, generated using the Vanderbilt USPP code. The relevant part of the methods section now reads:

“This was achieved by modifying the Carbon pseudopotential to generate a $1s^1$ core-hole, where both the ground-state and core-excited state pseudopotentials were prepared using ultrasoft Vanderbilt pseudopotential scheme.⁴⁹”

How many empty bands were included in the XCH calculations, and was this parameter converged/benchmarked?

Empty bands at least double the amount of occupied band were included in the NSCF calculation, which we verified was enough to fully converge the spectra. The relevant part of the methods section now reads:

“Empty bands at least double the amount of occupied band were included in the NSCF calculation.”

Was the full-core-hole approximation used?

The reviewer is correct and there was a typo in our original submission. The ShirleyXAS + MBXAS scheme operates within the full core hole approximation. We have corrected this mistake in the current manuscript and the methods section now reads:

“We simulated the Carbon K-edge XAS of graphite and Li-GICs using a many-body Δ SCF approach, ShirleyXAS⁴⁸ + MBXASPY.²⁷ The final electronic state due to the x-ray excitation was obtained using full core-hole approach, where the Carbon 1s electron was removed from the inner shell.”

Were the spectra post-processed, e.g., were they broadened or shifted?

We thank the reviewer for this question, which we agree was not adequately addressed in the original manuscript. Each spectrum was post-processed, including both at 0.1 eV numerical broadening and a 285.5 eV shift. The rigid shift is necessary due to our use of pseudopotentials, which unlike an all-electron calculation does not have an absolute reference energy. Thus, we independently calculated the shift necessary for aligning the simulated Carbon K-edge XAS of molecular CO to experiments, and used the same 285.5 eV shift throughout, which resulted in a prediction of the π^ peak in graphite to within*

0.2 eV of experiments. Our previous work employing this scheme within the excited core-hole approximation [Pascal, et al. JCP 140, 034107 (2014)], showed this approach to be robust for predicting the experimental edge of various Lithium solids at a similar level of accuracy. We also applied a dilation factor of 1.05 to each spectrum, in order to reproduce the experimental π^*/σ^* splitting in graphite seen experimentally. This empirical dilation is required due to the well-known inability of DFT at the PBE/GGA level of theory to reproduce the correct unoccupied energy spectrum. The relevant part of the methods now reads:

"...core-excited states obtained from the Slater determinant. Each spectrum was broadened by a Gaussian smoothening of 0.1 eV and a rigid shift of 285.5 eV was applied to each spectrum, based on a reference calculation where we aligned the Carbon K-edge of molecular CO to experiment. Further energy calibration was performed according to the formation energy scheme.^{28,50} This allowed us to calculate accurate energy shifts between the various Li-GICs and thus compare the spectra of systems with different chemical environments. To overcome the underestimation of the band-gap (and concomitantly the band-width) at this level of DFT, we applied an empirical dilation of 5% to each spectrum to match the splitting between the σ^* and π^* features in the experimental graphite spectrum, and this same dilation factor is applied to all Li-GICs spectra."

4. The manuscripts contains several typos that a spell-/grammar-checker should have detected. Note that the name "Monkhorst" is misspelled.

This and other typos have been corrected.

Reviewer #2 (Remarks to the Author):

We thank the review for your review, and for finding out paper interesting. We address your concerns below. Your comments are reproduced verbatim in *italics* while our response follows. As noted in our response to reviewer #1, we will indicate any major changes to the manuscript or SI by underline.

This is a short, interesting paper that reports on the impact of lattice and ion dynamics in lithium intercalated graphite. The authors calculate the carbon K-edge X-ray absorption spectra (XAS) and compare their results to experiment and conclude that the vibrational properties have a strong impact on the thermodynamics of lithium intercalation. Well interesting, this paper is not significantly important to the field and there are some technical limitations that prevent its publication.

We thank the reviewer for their positive reception of our work.

The introduction is quite long relative to the length of the body of the paper. It seems that much in the introduction is unnecessary.

We have reworked the introduction based on additional comments from Reviewer #1, which we believe adds further clarity. We appreciate the point regarding the length of the introduction, but respectfully disagree. As pointed out by reviewer #1, there has been quite a bit of work in this space both on the experimental and computational sides, which we find necessary to summarize in order to place our new findings in the proper context.

In addition, the first paragraph of the introduction states that graphite can be used as a cathode and in fact uses this motivation throughout the rest of the paper. This is highly problematic as graphite in all lithium-ion batteries are anodes. The potential of lithiated graphite is very low potential (about 0.1 volts versus Li metal) and this would render any battery with graphite as a cathode basically useless in terms of energy density.

We thank the reviewer for pointing out this typo, which was commented on by reviewer #1 as well. We have fixed this and other issues throughout the manuscript.

The introduction needs to be condensed and corrected. In addition, the importance of the results are not adequately explained. For example, what is the impact of these on lithium intercalation on battery operation. The discussion in the conclusion related to phonon activation with electromagnetic fields seems highly unlikely given that lithium ion batteries all have metal casings which are required for current conduction from the electrodes to the external circuit.

We believe that understanding the mechanism and dynamics in this well studied, model system is of significant importance, as it provides a basis for more sophisticated attempt to at anode engineering. We agree that the original conclusion was somewhat speculative, and have added the following sentence related to the temperature dependence of the anharmonic dynamics, and any consequence it may have on battery performance:

“Beyond accounting for the effect of temperature on performance, we propose that these effects opens up new avenues for non-equilibrium manipulation of materials, such as the recently demonstrated effects of light illumination⁴¹ or by selective phonon activation. Finally, the role of dynamics, and the associated “configurational entropy” has recently been proposed as critical for understanding the

cathode materials in Lithium-ion batteries.⁴² Our work expands on these efforts and presents a general scheme for further understanding the subtle physics that underly the function of more complex cells.”

The authors make strong statements about their room-temperature XAS results agreeing with the experiment as opposed to the OK results (no lattice dynamics); this is based on figure 1C which just looks at the peak areas and may not be representative of the complete XAS spectra. The authors need to directly compare the calculated XAS with those of the experiments in order to show that their simulations provide more accurate results. Similar comments apply to the Raman results in Figure 4. These comparisons are essential.

This is an excellent point, and we agree with the reviewer that this was not adequately presented in the original manuscript. We have now included in the supplementary materials (new Figure S1) a plot of the entire spectrum, comparing the experiment, OK simulated and 298K simulated spectra. As can be seen, the agreement in the XAS between the experiments and the finite-temperature simulation extends beyond the 1st peak, and includes the higher energy lineshapes as well, while the OK simulated XAS shows significant deviation.

On Page 3 in the second full paragraph, the authors state that there is incomplete charge transfer to the lithium ions and that the formal charge on the lithium is +0.8, +0.7 and +0.6 for stages I to III. It is unclear how this could possibly agree with the experiment which largely sees complete charge transfer to the Li and is the basis for the operation of Li-ion batteries (i.e., charge capacity). These results need to be rationalized and explained.

We thank the reviewer for this comment. We note however that the experimental energy capacity is related solely to the following elementary reaction at the anode:

The only requirement is a neutral Li-graphite electrode, which says nothing about the distribution of the charges on the Li and C atoms upon intercalation (only that they need to balance). We have added the following sentence to the manuscript discussion to address the issue of possible incomplete charge transfer:

“... in stage I, II and III respectively. This is in general agreement with a previous DFT study demonstrating a similar degree of charge transfer in Li-GICs, arising from dominant interactions and hybridization between Li 2s and C 2p_z states.³¹”

The last paragraph on page 2 discusses what aspects of the intercalated graphite structure give rise to the XAS features. The conclusions for this are obvious as XAS is a highly local probe and hence will only see the ordering within the planes (short bond lengths) and will be insensitive to the interactions between the graphite planes (longer distances). This paragraph adds very little to the paper that should be a move to the SI.

We thank the reviewer for this comment, however we believe that our discussion of the structural aspects that give rise to the observed spectral changes is important to explicitly state here, as it has not been elaborated previously, and lays the foundation for our future discussion of the effect of lithium dynamics and charge fluctuations.

Reviewer #3 (Remarks to the Author):

We thank the reviewer for your thorough review, and comments, which we address below, and we believe strengthens our paper. Your comments are reproduced verbatim in *italics* while our response follows.

The authors present a calculation-driven investigation of the motion of Li atoms intercalated into graphite, a system used as a cathode in lithium ion batteries. Not unexpectedly, they show that the lithium atoms move a fair bit, but more interestingly they also show that the distribution is very far from Gaussian (not reflective of a harmonic potential) and that this motion and its non-harmonic behavior has a dramatic effect on the enthalpy of the intercalated system. The predicted dynamics of the Li-graphite systems are fed in to first-principles calculations of the C K-edge x-ray absorption. As a point of comparison with experimental measurements, the peak ratio between the two largest features in the XAS are compared as a function of Li concentration (staging), and the authors find very good agreement between the predictions and measurements for this ratio.

We thank the reviewer for their summary, which captures the essence of the work. We note that our work mostly concerns the entropy (not enthalpy) of the Li ions

I have a few major concerns. The comparison with experiment is limited to the ratio of the two main peaks in the C K edge. As the current paper is written, the authors hinge much of their arguments on the agreement between experiment and calculation of this ratio, but I do not think it is adequately supported.

1. Peak ratios are going to be sensitive to the details of the fitting, such as the determination of the background level and bounds of the peaks. The method needs to be clearly laid out in either the paper or the supplemental. I would suggest a supplemental figure illustrating this for both the calculated and measured spectra. It is not clear which experimental data was used for generating the comparison in Fig 1c, but it should be cited in the caption and in the text where Fig 1c is mentioned.

Thank you. We have included a section in the SI on the peak area fitting procedure and have presented the various parameters used in new Table S1 and Figure S2. We have also added the reference for experimental data in Fig 1c.

2. Are the qualitative changes in the calculated spectra that are evident in Fig 1b bourn out in experiment? Or changes in the onset energy of the sigma star?

We thank the reviewer for this comment. There is indeed modulation in the σ^* with increasing lithiation. In the experiment, the σ^* feature broadens as the Li concentration increases. Computationally, we also observe this behavior. Therefore, we have added a sentence in the manuscript clarifying this point:

"...for graphite and all three Li-GICs, both in peak positions and lineshape. Specifically, at the σ^* resonance, we observe broadening of this peak as staging progresses, in agreement with experiments. In contrast, the simulated XAS using the OK optimized structure (Fig. S1) was found to be in poor agreement with experiment, with significantly shaper peaks and generally overestimated oscillator strengths."

I have no major criticisms with the computational methodology used. In a few cases the authors need to be more clear. In particular:

3. What were the volumes used for each of the 4 systems?

The cell lattice parameters are shown in Fig. 1a in the main text and are now also tabulated as a new Table S1 in the supplementary materials. We have added the following to the start of the results section:

“The starting crystal structures of graphite and stage I – III Li-GICs are shown in Fig. 1a and presented in Table S1 of supplementary materials.”

Was this from experiment or from relaxing with respect to the pressure of the unit cell? If the latter, how does this compare with measured data, and if the former, what was the pressure from the DFT?

The initial structures are the experimental one, obtained from the Inorganic Crystal Structure database and relaxed and equilibrated in DFT at 298K. We have expanded the methods section to include these and other details following the comments from Reviewer #2 as follows:

“We resolved the temperature related stress in the system by means of a 2ps constant temperature, constant pressure (NPT) simulation. This was then followed by at least 10 ps of constant temperature, constant volume (NVT) dynamics.”

4. How many snapshots went into the XAS calculations? (you state the corresponding numbers for the partial charge analysis and vibration/entropy parts.)

We used a total of 5 snapshots from the trajectory (using every available C atom) to generate statistical average spectra.

What is meant by “statistical averaging” for generating the final spectra?

We have now included shaded bars in the spectra in the SI (Fig. S9) to indicate the uncertainty. We have added a sentence clarifying this point into the methodology section:

“For each MD snapshot, we simulated the XAS of each carbon atom individually, and the final spectra were obtained from statistical averaging. The reported results were obtained from averaging 5 uncorrelated snapshots over 5 ps, each 1 ps apart. The uncertainty in the simulated spectra due to statistical averaging is shown in Fig. S9.”

Additional comments:

Stylistically, the authors may want to put more weight into their finding that the Li entropy depends strongly on the anharmonic effects and the motion of the graphite sheets. The first full paragraph on page 2 mentions the entropy calculations, but goes on to only summarize XAS results. Along these lines, I think more emphasis could be placed on the important role that non-Gaussian dynamics can have on a system’s thermodynamics. The authors cover this well in the final paragraph before the methods, but it could be brought up in the introductory remarks as well.

We thank the reviewer for this comment. The anharmonic nature of the Lithium dynamics is indeed the

major point of this work, as it underlies the system thermodynamics and XAS spectra. We have made the following changes to the manuscript to emphasize this:

in the Results section:

“Thus overall, we show that it is crucial to explicitly consider both the harmonic and the anharmonic (i.e., the high population of states near the tail of the Lorentzian distribution) vibrational entropy, in order to accurately capture the thermodynamics of Li-GICs. Further, we predict that these anharmonic effects may be a general feature of Lithium intercalation compounds, which will be the subject of future studies.”

and in the Concluding paragraph:

“Finally, the role of dynamics, and the associated “configurational entropy” has recently been proposed as critical for understanding the cathode materials in Lithium-ion batteries.⁴² Our work expands on these efforts and presents a general scheme for further understanding the subtle physics that underly the function of more complex cells.”

The manuscript makes a distinction between XAS and XRS, but for low momentum transfer XRS can provide the same information. That is to say that, Ref 15 provides information directly comparable to the XAS calculations, and the techniques used here are perfectly fine for simulating low-q XRS.

We agree with the reviewer on this point, and have modified the text to now read:

“Of note, recently XRS, which at low momentum transfer produces information comparable to XAS, was used to provide insight into the staging and mechanism of Li intercalation in Li-GICs.²⁵”

The authors link the XAS to the motion of the Li through their AIMD. Would structural probes like neutrons be able to also confirm the level of in-plane and out-of-plane displacement (though perhaps limited to modeling only the width of the distribution)?

This is an excellent point, and we thank the reviewer for bringing this to our attention. Neutron scattering and other imaging probe have been used to show the evolution of the lithiation/delithiation process in LiC_x materials. However, to our knowledge, it does not provide information on the displacement of the Li within the graphite cage that we now show is possible to obtain from XAS.

We have made the following changes to the manuscript to detail for this:

“Previous works have utilized structural probes to track the motion of lithium ions in electrode materials during charge and discharge,³⁸⁻⁴⁰ and we expect that our predicted asymmetric ion distributions (in- and out-of-plane) with staging may be visible in these complementary techniques, particularly in high resolution neutron scattering.”

Fig 2a-c should be described better in the caption — what the color bar represents.

Thank you. We have updated caption to be clearer. It now reads:

“Figure 2 | Structural analysis of Li-GICs. a – c. Li atom librational displacement within the graphitic cage in LiC₆, LiC₁₂ and LiC₁₈ respectively showing the probability distribution function of Li positions”

within the graphitic cage. d. *In-plane* projection of the Li librational cage displacement dynamics. e. *Out-of-plane* projection of the Li librational cage displacement dynamics.”

In Fig 2d / AIMD trajectories are there any meaningful angle correlations for the Li motion with respect to the C hexagons?

We thank the reviewer for bringing up the discussion on angle correlation for the Li motion. We analyzed angles formed by the C hexagons and Li as the center atom and found that the distribution of the angle is relatively asymmetric. These results are presented in a new Figure S7. We have added a discussion of this part into the manuscript.

“...the distributions have large Kurtosis and are in fact Lorentzian. We also analyzed the angle correlation between the center of mass of the top hexagon, the Li atom position and the center of mass of the bottom hexagon, which shows similar non-gaussian distributions (Fig. S7).”

Optional, but would it be feasible to also include calculations of the vibrational DOS from within an entirely harmonic approximation? This might give an interesting point of comparison for harmonic vs anharmonic motion.

We thank the review for the comment. The anharmonicity from our approach does not arise from the method employed but due to the motion itself being anharmonic. The vibrational DOS is approximated using a quantum harmonic oscillator (QHO) weighting function, assuming that each of the individual modes are independent. As we point out in the text, if one uses the vibrational modes that may come out of a 0K phonon QM calculation, this will lead to a constant value of $\Delta S = -7.6$ J/mol/K for all Lithium stages. Thus one needs to sample an anharmonic distribution (as we obtained from our AIMD) in order to fully capture the physics. We have this added the following clarifying statement to the methods section:

“...where the entropy is calculated from the velocity autocorrelation function.^{35,36} By adequately sampling the non-gaussian ion distribution, our approach explicitly captures both harmonic and anharmonic effects.”

as well as to the methods section:

“...then we calculated the entropy by separately considering the diffusive and vibrational motions of the atoms and integrating with the appropriate quantum weighting functions. Notably, since the vDoS is obtained directly from MD, the resulting entropy inherently contains contributions due to harmonic (purely vibrational), as well as anharmonic (librational) and self-diffusive motions. We find that the latter two are the significant contributions to the Li entropy.”

REVIEWER COMMENTS

Reviewer #1 (Remarks to the Author):

In their revision, the authors added methodological details and a brief discussion of the prior computational work in response to my questions. However, some of these details along with the responses to the other reviewers raise additional questions that should be addressed in a second revision.

1. Convergence of the XAS simulations

In the revised manuscript, qualitative statements regarding the numerical convergence of the simulations are made, but no quantitative evidence is shown. The authors now mention that "at least twice as many" empty bands than occupied bands were used in their calculations. Please report an example of the DOS convergence with the number of bands in the relevant energy range as supporting information. Similarly, the convergence of the XAS simulation with the cell size needs to be demonstrated at least for one example (also in the SI). The band and cell-size convergence are critically important to obtain correct peak ratios, which is at the core of the present work.

2. Minor comment: Li metal batteries are not common

In the revised introduction the following statement is made: "Modern cathodes usually comprise transition metal oxides, with anodes comprised of, most recently, Lithium metal." This is incorrect. The battery community typically distinguishes between lithium-ion batteries (LIBs, e.g., graphite-based anodes) and lithium-metal batteries (LIMs, Li metal anodes). Li metal is used only in very few commercial batteries.

Reviewer #2 (Remarks to the Author):

This paper has been revised with respect to the comments from the referees, and while the paper is improved, the revisions have brought up some new concerns and the importance and relevance of the paper are still not well explained. It is recommended that after revision the paper should be submitted to a more specialized journal.

While the introduction now correctly associates the anode with graphite, there are statements in the revision that are not correct. Lithium metal is not really used in any commercial batteries and is still very much under development. In addition, Li-ion battery anodes are not highly oriented pyrolytic graphite (HOPG) but rather consist of rather specialized graphite particles that are embedded in appropriate binder. This needs to be corrected.

The authors have now compared their results with the simulations in figure S1. However, it is not clear how these XAS spectra are normalized. In experiments XAS is normalized so that the intensity at high energies goes to unity or the XAS is on a per atom basis. In Figure S1, it appears as if the XAS spectra are normalized at the π^* peak near 286 eV. This makes a comparison with experiments uncertain. Further, based on Figure S1, it is not clear that the authors model is much better than the static model as both miss aspects of peak heights and positions. The authors need to explain wider their dynamic model simulation is better than the static model using some kind of objective metric.

The importance of the paper is still not clearly articulated and hence a more specialized journal is appropriate. What is the impact of the work on the broader electrochemical energy storage community? How can this insight be used to design new materials or better understand the existing graphite anodes? While the paper is a good contribution, the case for publication in Nature Communications is not adequately made.

Reviewer #3 (Remarks to the Author):

The authors have revised their manuscript. I believe they have addresses my questions and comments as well as those from the other reviewers. I recommend that this manuscript be accepted with only very minor corrections.

The phrase "while also reducing the self-interaction between atoms" in the methods section might be better said along the lines of "while also reducing the effect of artificial periodicity". My concern with self-interaction is that might make the reader think about self-interaction errors in mean-field theories like DFT.

In the methods section NSCF is used without being first defined. Here it might be better to just specify that these bands were used to calculated the excited states for the XAS instead of going in to the use of SCF vs NSCF for X-ray calculations.

RESPONSE TO REVIEWERS' COMMENTS

Reviewer #1 (Remarks to the Author):

We thank the reviewer for your review of our work. Your comments were very helpful in improving the overall quality the manuscript. We address your concerns below. Your comments are reproduced verbatim in *italics* while our response follows in red. Where appropriate, we will indicate any major changes to the manuscript or SI by underline.

In their revision, the authors added methodological details and a brief discussion of the prior computational work in response to my questions. However, some of these details along with the responses to the other reviewers raise additional questions that should be addressed in a second revision.

1. Convergence of the XAS simulations

In the revised manuscript, qualitative statements regarding the numerical convergence of the simulations are made, but no quantitative evidence is shown. The authors now mention that "at least twice as many" empty bands than occupied bands were used in their calculations. Please report an example of the DOS convergence with the number of bands in the relevant energy range as supporting information. Similarly, the convergence of the XAS simulation with the cell size needs to be demonstrated at least for one example (also in the SI). The band and cell-size convergence are critically important to obtain correct peak ratios, which is at the core of the present work.

We thank the reviewer for the question. We have included the convergence of the XAS simulation as a function of supercell size and number of empty bands. The manuscript now reads:

"included in the NSCF calculation. The convergences of XAS simulation as function of supercell size or number of empty bands are shown in Fig. S9-10 where we find that the convergence is achieved at 3x3x3 unit cell with minimum band factor of 3."

2. Minor comment: Li metal batteries are not common

In the revised introduction the following statement is made: "Modern cathodes usually comprise transition metal oxides, with anodes comprised of, most recently, Lithium metal." This is incorrect. The battery community typically distinguishes between lithium-ion batteries (LIBs, e.g., graphite-based anodes) and lithium-metal batteries (LIMs, Li metal anodes). Li metal is used only in very few commercial batteries.

We thank the reviewer for this comment. We have reworked the intro to now read:

"Modern, commercial LiB cathodes usually comprise transition metal oxides,⁶ with anodes comprised of synthetic graphite, both of which are integrated in composites together with amorphous conductive carbon and polymer binders. During the electrochemical cycling of LIBs, these graphite anodes undergo the reversible lithiation and delithiation, dynamically forming and collapsing lithium-graphite intercalation compounds (Li-GICs) with various degrees of staging.⁶"

Reviewer #2 (Remarks to the Author):

We thank the reviewer for taking the time to review of our work. We address your concerns below. Your comments are reproduced verbatim in *italics* while our response follows in red. Where appropriate, we will indicate any major changes to the manuscript or SI by underline.

This paper has been revised with respect to the comments from the referees, and while the paper is improved, the revisions have brought up some new concerns and the importance and relevance of the paper are still not well explained. It is recommended that after revision the paper should be submitted to a more specialized journal.

While the introduction now correctly associates the anode with graphite, there are statements in the revision that are not correct. Lithium metal is not really used in any commercial batteries and is still very much under development. In addition, Li-ion battery anodes are not highly oriented pyrolytic graphite (HOPG) but rather consist of rather specialized graphite particles that are embedded in appropriate binder. This needs to be corrected.

We apologize for this error, as was pointed out by reviewer #1 as well. We have reworked the intro to now read:

“Modern, commercial LiB cathodes usually comprise transition metal oxides,⁶ with anodes comprised of synthetic graphite, both of which are integrated in composites together with amorphous conductive carbon and polymer binders. During the electrochemical cycling of LiBs, these graphite anodes undergo the reversible lithiation and delithiation, dynamically forming and collapsing lithium-graphite intercalation compounds (Li-GICs) with various degrees of staging.⁶”

The authors have now compared their results with the simulations in figure S1. However, it is not clear how these XAS spectra are normalized. In experiments XAS is normalized so that the intensity at high energies goes to unity or the XAS is on a per atom basis. In Figure S1, it appears as if the XAS spectra are normalized at the π^* peak near 286 eV. This makes a comparison with experiments uncertain.

The spectra are normalized according to the π^* peak area under the curve from the Lorentzian fit. The normalizing factors are used to scale the spectra such that the area under the curves are equal across all the data we presented. We have added the detail to the XAS methodology that now reads:

“The spectra are normalized according to the area under the curve from the Lorentzian fit at π^* excitation to be equal across all the spectra”

Further, based on Figure S1, it is not clear that the authors model is much better than the static model as both miss aspects of peak heights and positions. The authors need to explain wider their dynamic model simulation is better than the static model using some kind of objective metric.

As a figure of merit, in Figure 1c and Table S1, we present the peak area ratios, showing that the dynamic model significantly improves the agreement with experiment comparison to the static model.

The importance of the paper is still not clearly articulated and hence a more specialized journal is appropriate. What is the impact of the work on the broader electrochemical energy storage community?

How can this insight be used to design new materials or better understand the existing graphite anodes? While the paper is a good contribution, the case for publication in Nature Communications is not adequately made.

EELS/XAS is widely used to study the electronic states of intercalated Lithium battery materials, and Lithiated graphite represents a fundamental test of our understanding of electrochemical systems [Gasteiger et al, JACS 2011, 133, 120502]. In this paper, we quantify, for the first time, the role of dynamics in modulating the electronic states in a model system. Atomic dynamics inform the thermodynamic states, and we have developed an approach that allows for the prediction of subtle entropy changes in excellent agreement with experiments (Fig. 4a), which can be easily extended to more complex material systems and represents a significant improvement on traditional approaches. This is summarized in the main text:

“These vibrational modes are then shown to be critical for accurately accounting for the system thermodynamics, beyond what can be predicted from a purely harmonic theory, and thus resolves some of the disagreement between previous calculations and experiments that exists in the literature.”

Reviewer #3 (Remarks to the Author):

We thank the reviewer for your review of our work. Below we address your concerns, which were helpful in improving the overall quality of our work. Your comments are reproduced verbatim in *italics* while our response follows in red. Where appropriate, we will indicate any major changes to the manuscript or SI by underline.

The authors have revised their manuscript. I believe they have addresses my questions and comments as well as those from the other reviewers. I recommend that this manuscript be accepted with only very minor corrections.

The phrase "while also reducing the self-interaction between atoms" in the methods section might be better said along the lines of "while also reducing the effect of artificial periodicity". My concern with self-interaction is that might make the reader think about self-interaction errors in mean-field theories like DFT.

We thanks the reviewer for the suggestion on clarifying the method section. We have modified the part to now read:

"...while also reducing the self-interaction between atoms due to the effect of artificial periodicity"

In the methods section NSCF is used without being first defined. Here it might be better to just specify that these bands were used to calculated the excited states for the XAS instead of going in to the use of SCF vs NSCF for X-ray calculations.

We thanks the reviewer for the comment. We have added a sentence addressing the NSCF calculation. The manuscript now reads:

"Next, we performed non self-consistency field calculation (NSCF), a single shot calculation to construct the Hamiltonian without updating the charge density, to access empty states at higher energy"

REVIEWER COMMENTS

Reviewer #1 (Remarks to the Author):

In this revision, the authors have addressed all of my remaining concerns. I can now recommend the manuscript for publication.

Reviewer #2 (Remarks to the Author):

This paper has been improved; however, I do not believe edit that in its present form it is publishable in Nature Communications. The authors main point is it their simulated XAS gives better agreement when incorporating dynamics then previous results. The shown data do not appear to support this conclusion. All experimental XAS data, as pointed out in the previous review, are normalized at the highest energy and thus it is not a fair or valid comparison when the authors normalize under the π^* peak. This renders any comparison uncertain. In addition, the authors state that table S1 shows the dynamical model significantly improves the agreement with experiment. Table S1, in fact, contains no results from previous simulations that do not treat dynamics. Thus, it is impossible to come to this conclusion. In addition due to the normalization, it is also impossible to make any comparison between the authors simulations and the experiment.

Two minor points: first in graphite anodes, the graphite is not synthetic but mined. second in table S1 and all the tables the authors need to pay attention to significant figures.

RESPONSE TO REVIEWERS' COMMENTS

Reviewer #1 (Remarks to the Author):

In this revision, the authors have addressed all of my remaining concerns. I can now recommend the manuscript for publication.

We are delighted that the reviewer is supportive of publication and will like to thank them for their critical comments.

Reviewer #2 (Remarks to the Author):

We thank the reviewer for your review of our work. We address your concerns below. Your comments are reproduced verbatim in *italics* while our response follows in red. Where appropriate, we will indicate any changes to the manuscript of SI by underline.

This paper has been improved; however, I do not believe edit that in its present form it is publishable in Nature Communications. The authors main point is it their simulated XAS gives better agreement when incorporating dynamics then previous results. The shown data do not appear to support this conclusion. All experimental XAS data, as pointed out in the previous review, are normalized at the highest energy and thus it is not a fair or valid comparison when the authors normalize under the π^ peak. This renders any comparison uncertain. In addition, the authors state that table S1 shows the dynamical model significantly improves the agreement with experiment. Table S1, in fact, contains no results from previous simulations that do not treat dynamics. Thus, it is impossible to come to this conclusion. In addition due to the normalization, it is also impossible to make any comparison between the authors simulations and the experiment.*

We thank the reviewer for this question, regarding the (dis)agreement between the simulated XAS based on finite temperature MD sampling vs that of the static crystal, compared to experiment. The reviewer is correct in that different normalization factors are used in experiment and theory. However we believe that given the uncertainty in the peak intensities at higher energies (in both cases – elaborated below), normalizing the theory to the high energy scattering region would not be appropriate (and is in fact not commonly done in practice), and that our approach is justified.

First, we note that in the experiments, XAS spectra background subtraction is generally performed from a line regression analysis, starting at the pre-edge region with a polynomial regression to the post-edge region. This approach, while standardly in the community, introduces some uncertainty, and sometimes results in a “flattening” in the relative intensity of the post-edge features. On the other hand, in our simulations there are some uncertainties in the peak intensities of the higher energy features, since in the Δ SCF approach employed in the current study, excited states beyond the main-edge are generated non self-consistently, assuming the same potential as the 1st excited state. Our experience has been that these lead to a loss of intensity in features further away from the main-edge, as these states become increasingly Rydberg-like in nature and are more difficult to model correctly with DFT [see our recent work with the Cabana group on XAS of MoS₂ – doi://10.26434/chemrxiv-2022-zpl7n]. Moreover, the raw data from our calculations integrate to unity, representing the probability of observing an excited electron over the specified energy range. This latter point thus gives us an unambiguous way of comparing all our spectra: we match the intensity of the π^* peak of graphite in our calculations with that of the experiment and applied the same scaling factor for all other simulated spectra.

We have modified the manuscript to further elaborate this point as followed:

“... to statistical averaging is shown in Fig. S11. We then matched the π^* peak intensity in graphite from experiments to that of our simulated XAS of graphite and used the same scaling factors for all other simulated spectra. This was necessary due to the inherent differences in the normalization methods between the experiment and simulation. In the experiment, the spectra are normalized based on background signal subtraction whereas the calculation does not have any background response. Nevertheless, these differences are self-contained which allow us to quantitatively compare spectra changes across staging such as correctly predicting the area ratio.”

To the second point, we have now included the comparison of the Lorentzian fitting of the peaks between the static crystal structures and the average structure from MD in Table S1. This data was used to create Figure 1c, which clearly shows that the improved performance of the MD sampled XAS in reproducing the experimental XAS of the various Li-GICs, reproduced below:

Rebuttal Figure R1: (Figure 1c in main text): Comparison of the π^*/σ^* peak area ratio for the various Li-GICs, from experiment from Boesenberg14 (blue circles), simulations of the static OK optimized structure (red triangles) and simulations sampled from a 298K AIMD trajectory (orange squares)

Two minor points: first in graphite anodes, the graphite is not synthetic but mined.

Thank you. We have removed the work "synthetic" from the main text.

second in table S1 and all the tables the authors need to pay attention to significant figures.

We adjusted the significant figures in Table S1 and in all supplementary tables.

REVIEWERS' COMMENTS

Reviewer #2 (Remarks to the Author):

the authors have addressed my concerns & the paper can be accepted. Figure 1c is convincing.

REVIEWERS' COMMENTS

Reviewer #2 (Remarks to the Author):

the authors have addressed my concerns & the paper can be accepted. Figure 1c is convincing.

We are delighted that the reviewer is supportive of publication and will like to thank them for their comprehensive review.